# Local Consolidative Therapy for Oligometastatic Non-Small Cell Lung Cancer

**DOI:** 10.3390/cancers14163977

**Published:** 2022-08-17

**Authors:** Patricia Mae G. Santos, Xingzhe Li, Daniel R. Gomez

**Affiliations:** Department of Radiation Oncology, Memorial Sloan Kettering Cancer Center, New York, NY 10065, USA

**Keywords:** local ablative therapy (LAT), lung cancer, metastatic disease, non-small cell lung cancer (NSCLC), oligometastatic disease, oligoprogression, stereotactic ablative therapy (SABR), stereotactic body radiotherapy (SBRT)

## Abstract

**Simple Summary:**

Metastatic non-small cell lung cancer (NSCLC) classically portends a poor prognosis and worse overall survival. However, recent advances in modern systemic therapy and the increasing recognition of a distant clinical entity known as “oligometastatic disease”—i.e., a controlled primary tumor and a limited number of distant lesions (≤5 metastases)—have led to paradigm shifts in management. Findings from Phase II randomized clinical trials suggest that aggressive local consolidative therapy (LAT) in the form of surgery or highly conformal radiation, known as stereotactic ablative body radiotherapy (SABR), may help to significantly mitigate disease progression and prolong survival. In this review, we summarize clinical evidence from published and ongoing trials that support the use of LAT/SABR in the treatment of oligometastatic NSCLC. We discuss key findings and caveats to these studies, and we highlight potential considerations and avenues for further investigation in the oligometastatic disease space.

**Abstract:**

In the last 20 years, significant strides have been made in our understanding of the biological mechanisms driving disease pathogenesis in metastatic non-small cell lung cancer (NSCLC). Notably, the development and application of predictive biomarkers as well as refined treatment regimens in the form of chemoimmunotherapy and novel targeted agents have led to substantial improvements in survival. Parallel to these remarkable advancements in modern systemic therapy has been a growing recognition of “oligometastatic disease” as a distinct clinical entity—defined by the presence of a controlled primary tumor and ≤5 sites of metastatic disease amenable to local consolidative therapy (LAT), with surgery or stereotactic ablative body radiotherapy (SABR). To date, three randomized studies have provided clinical evidence supporting the use of LAT/SABR in the treatment of oligometastatic NSCLC. In this review, we summarize clinical evidence from these landmark studies and highlight ongoing trials evaluating the use of LAT/SABR in a variety of clinical contexts along the oligometastatic disease spectrum. We discuss important implications and caveats of the available data, including considerations surrounding patient selection and application in routine clinical practice. We conclude by offering potential avenues for further investigation in the oligometastatic disease space.

## 1. Introduction

Lung cancer is the third most prevalent cancer diagnosis in the United States [1] and the leading cause of cancer-related mortality in the world, accounting for 1.76 million deaths annually [2]. For patients with localized disease (Stages I-III), the aggregate 5-year survival rate is roughly 60 percent [3]. In contrast, prognosis in patients with Stage IV disease is exceptionally poor, with an estimated survival of only 8% at 5 years [4]. Furthermore, of the approximately 2 million cases of lung cancer diagnosed each year, over half will have metastatic disease at presentation. Despite this dismal outlook, significant strides have been made in our understanding of the underlying biological mechanisms driving disease pathogenesis. This is particularly true for non-small cell lung cancer (NSCLC), a group of histologic subtypes that account for approximately 85% of all lung cancer diagnoses [5]. In the past decade alone, the discovery and application of important predictive biomarkers, along with advancements in systemic therapies, have opened new avenues for the treatment of metastatic NSCLC.

Historically, patients with Stage IV NSCLC were treated indiscriminately with cytotoxic chemotherapy, which offered a median progression-free (PFS) and overall survival (OS) benefit of only 4 to 6 months and 8 to 14 months, respectively [6,7,8]. Since then, several landmark trials published between 2006 and 2020 have led to the broader use of chemoimmunotherapy regimens. Specifically, the addition of single-agent PD-1 inhibitors such as pembrolizumab and atezolizumab to platinum doublet chemotherapy led to modest but notable improvements in oncologic outcomes (i.e., median PFS, 7 to 9 months and median OS, 17 to 22 months) [9,10,11,12]. Among patients with targetable gene alterations, the survival benefit has been even more impressive. In the FLAURA study, first-line treatment of *EGFR* (exon 19 deletion or *L858R*) mutant NSCLC with the third generation tyrosine kinase inhibitor osimertinib resulted in a significant improvement in median PFS and OS of 19 months and 39 months, respectively [13,14]. Similarly, in the ALEX trial, alectinib for *ALK* rearrangement of positive NSCLC was associated with a median PFS of 35 months [15]. As such, immunohistochemical staining with *PD-L1* along with molecular testing for *ALK* rearrangements, *EGFR* or *KRAS* mutations, and other emerging oncogenic drivers is now part of the standard of care.

Concomitant with advancements in modern systemic therapy for metastatic NSCLC, has been a parallel rise in the use of “ablative” techniques as local consolidative or local ablative therapy (LAT) for the treatment of limited or oligometastatic disease. Classically defined by the presence of five metastases or less [16], the origins of the “oligometastatic hypothesis” date back to the 1960s through the 1980s, when prominent physicians posited whether metastasis-directed surgical and radiotherapeutic interventions could be potentially curative in select patients with limited metastatic disease [17]. However, it was not until 1995 that the term “oligometastases” was coined by Dr. Samuel Hellman and Dr. Ralph Weichselbaum in their seminal editorial published in the *Journal of Clinical Oncology* [16]. Now, over 25 years since the introduction of the “oligometastatic hypothesis,” the existence of an intermediary state between localized cancer and widely metastatic disease remains a matter of ongoing scientific discussion and fervent debate [18,19,20].

Nevertheless, intriguing findings from recent phase II randomized trials [21,22,23,24,25] have led to both (1) the growing recognition of the oligometastatic state as its own distinct clinical entity from “poly-“ or widespread metastatic disease, and (2) the increasing adoption of ablative techniques, such as stereotactic ablative radiotherapy (SABR), interventional-radiology guided ablation, or surgery, collectively termed local ablative therapy (LAT), in the treatment of oligometastatic disease. For patients with Stage IV NSCLC, the implications of this shifting paradigm are particularly salient, as approximately 25 to 50 percent of patients are estimated to have oligometastatic disease at initial diagnosis [26,27,28]. However, as Phase III data have yet to mature, many questions still remain regarding appropriate use.

In this article, we will review key findings from Phase II trials of LAT in oligometastatic NSCLC, along with emerging data from ongoing Phase III randomized controlled studies. We will discuss questions surrounding the application of LAT in various clinical contexts, including oligoprogressive disease. We will highlight special considerations when using LAT or ablative therapies in vulnerable patient populations (e.g., older adults). Lastly, we will discuss future opportunities to achieve therapeutic personalization, including efforts to identify prognostic factors, including biomarkers to predict and assess response to treatment.

## 2. Defining the Oligometastatic Disease State

### Synchronous Versus Metachronous Oligometastatic Disease

In a recent ESTRO-ASTRO consensus statement published in 2020, Lievens et al. [29] defined the oligometastatic disease state primarily by the existence of 1–5 metastatic lesions, largely independent of location, provided that they are all safely treatable (Table 1). Notably, primary tumor control is considered optional [29]. As such, oligometastatic disease can be defined as either synchronous or metachronous with diagnosis of the primary tumor. Synchronous oligometastatic disease is detected at the time of and up to 3 months after initial diagnosis, with the patient having an untreated primary tumor and limited metastases simultaneously [29]. In contrast, metachronous oligometastatic disease (sometimes referred to as “oligorecurrent” disease) is defined as the detection of limited metastases at least 3 months after the initial diagnosis and often after control of the primary tumor is achieved after treatment with curative intent [29].

Importantly, the authors noted that the distinction between synchronous versus metachronous oligometastatic disease is critical as some studies report a better prognosis among patients with metachronous oligometastatic disease [30,31], though this has not been consistently observed [32,33]. Additionally, while there are some concerns that shorter disease-free intervals between the attainment of primary tumor control and subsequent development of metachronous oligometastatic disease may also be of prognostic significance [34,35,36], limited data preclude a consensus regarding its importance.

## 3. Clinical Data Supporting LAT in Oligometastatic NSCLC

### 3.1. Retrospective Studies

In the absence of randomized evidence, several large, single institutional retrospective series first attempted to provide data supporting the safety and efficacy of LAT in oligometastatic disease [23,24,25,26]. Most studies focused on SBRT/SABR. In one large individual patient meta-analysis of 757 patients with oligometastatic NSCLC, the authors reported outcomes of patients who received ablative therapy to all sites of metastatic disease. In this study, most underwent surgical metastasectomy (62%), while the remainder received some form of radiotherapy (38%). Median OS was 26 months, and the median PFS was 11 months. In the most favorable group by RPA analysis, patients with metachronous oligometastatic disease had an OS rate of 48% at 5 years. Overall, 98% of patients had ≤3 lesions, and 88% had a solitary lesion.

Using a propensity score-matched analysis from MD Anderson Cancer Center (MDACC), Sheu et al. [43] reported outcomes of 90 patients with synchronous oligometastases (≤3 sites) treated with at least two cycles of chemotherapy, followed by LAT with surgery or radiation compared with those who did not. In this study, the median OS was higher in patients who received LAT (27 vs. 13 months). Similarly, a pooled analysis from Germany of 700 patients with lung metastases (30% NSCLC) showed the benefit with lung SBRT, with a 2-year OS of 54% in the NSCLC subset [44]. Importantly, although encouraging, these data were limited by their retrospective nature and subject to immortal time bias, thus hindering generalizability.

### 3.2. Clinical Trials

Building on findings from prior retrospective series and a handful of small, single-arm prospective trials, three Phase II randomized studies provided the first high-level evidence to support LAT with SBRT/SABR for oligometastatic disease: (1) the MDACC study (Gomez et al. [22,24]), (2) the UT Southwestern study (Iyengar et al. [21]), and (3) the SABR-COMET trial (Palma et al. [23,25]). These studies are summarized in Table 2.

The earliest among these was the MDACC study, a multicenter, randomized, controlled trial of aggressive LAT versus maintenance therapy/observation (MT/O) alone for oligometastatic NSCLC first reported by Gomez et al. in 2016 [22]. Acceptable forms of LAT included surgery or radiation. Patients were eligible for study inclusion if they had ≤3 lesions and no disease progression after successful completion of first-line systemic therapy. Randomization was balanced based on several prognostic factors, including the number of metastases, response to initial chemotherapy, the presence of CNS involvement, intrathoracic nodal status, and *EGFR* or *ALK* mutational status. The primary study endpoint was PFS. Of note, although 74 patients were enrolled in the trial, the study closed early to accrual due to the significant PFS benefit seen on analysis by the Data Safety Monitoring Board. Of the 49 patients who did undergo randomization, patients who received aggressive LAT had improved PFS compared to MT/O alone (14.2 months [95% CI 7.4 to 23.1 months] vs. 4.4 months [95% CI, 2.2 to 8.3 months]; *p* = 0.022). In the 2019 update, the authors reported a significant OS benefit among patients who received LAT relative to MT/O (41.2 months [95% CI 18.9 to not reached] vs. 17.0 months [95% CI, 10.1 to 39.8 months]; *p* = 0.017), with a median follow-up of 38.8 months [22]. There were no significant differences in toxicity between the two study arms and no Grade ≥4 adverse events [22,24]

Then, in 2018, Iyengar et al. at UT Southwestern published a smaller, single institution, randomized trial of LAT with SABR and maintenance chemotherapy versus maintenance chemotherapy alone in oligometastatic NSCLC [21]. Patients were eligible for study inclusion if they had a controlled primary tumor plus up to five metastatic lesions. Patients were required to have either stable disease or at least a partial response to induction chemotherapy prior to enrollment. In contrast to the Gomez et al. study, patients were excluded if they had targetable *EGFR* mutations or *ALK* rearrangements. The primary study endpoint was PFS. Of the 29 patients enrolled in this study, those who received SABR in addition to maintenance chemotherapy had a longer median PFS compared to those who received maintenance chemotherapy alone (9.7 months versus 3.5 months; *p* = 0.01). Once again, this trial was also closed early to accrual due to the significant PFS benefit seen in the planned interim analysis. Toxicities were similar between the two study arms, with lower rates of local recurrence and no in-field failures with SABR.

Lastly, as the largest and most recent of these studies, the SABR-COMET trial was first published by Palma et al. in 2019 [23]. Patients were eligible for enrollment if they had a controlled primary tumor plus 1–5 metastatic lesions, had an EGOG performance status of 0–1, and had a life expectancy of at least 6 months. Unlike the preceding randomized trials by Gomez et al. [22,24] and Iyengar et al. [21], in which study participation was limited only to patients with NSCLC, the SABR-COMET trial enrolled patients with oligometastatic disease, regardless of their primary tumor histology. Overall, of the 99 patients ultimately enrolled in this multicenter, randomized, controlled trial, only 18% had an NSCLC primary; however, approximately 47% of the treated lesions were in the lungs. After initial stratification by the number of metastases (1–3 vs. 4–5), patients were randomized to standard of care (SOC) treatment plus SABR versus SOC treatment alone. The primary endpoint was overall survival.

In the initial SABR-COMET report published in 2019 [23], patients who received SABR in addition to the SOC had significantly higher median OS compared to those who received SOC therapy alone (41 months [95% CI, 26 to not reached] versus 28 months [95% CI, 19 to 33]; *p* = 0.09), with a median follow-up of 25 months (IQR, 19 to 54 months). In the most recent update published in 2020 [25], long-term results showed that the survival benefit was durable after a median follow-up of 51 months. Patients who received SABR plus SOC had a significantly improved OS rate of 42.3% (95% CI 28% to 56%) at 5 years versus only 17.7% (95% CI 6 to 34%) in patients who received SOC alone. Rates of Grade ≥2 toxicity were 20% more common in the SABR cohort (*p* = 0.026); however, there were no significant differences in QOL with long-term follow-up. Importantly, treatment-related deaths occurred in 3 out of 66 (4.5%) patients treated with SABR, of which two were secondary to treatment of oligometastatic disease in the lung.

Collectively, findings from these three randomized trials support the use of SABR as a generally safe and effective method of LAT for the treatment of oligometastatic NSCLC. Across studies, patients consistently derived significant benefits in terms of greater freedom from disease progression and prolonged survival—and for some, the potential for cure. Understandably, there has been growing excitement over the utilization of SABR in the oligometastatic setting, with some clinicians choosing to incorporate this treatment paradigm as part of routine practice. However, practicing radiation oncologists should interpret these findings with care as there are several important limitations to address before SABR can be considered as the preferred treatment for oligometastatic NSCLC. First, given the Phase II nature of these trials, these findings, though instructive, cannot yet be interpreted as definitive, as findings from Phase II data do not always translate to the larger Phase III setting. Second, owing to early closure in two out of three studies, along with the histology-agnostic methodology of the SABR-COMET trial, the sample sizes of oligometastatic NSCLC were considerably small. As such, further subset analysis of potential covariates of interest is difficult. Third, given that these trials were opened between 2011 and 2014, these findings pre-date the publication of several large immunotherapy and chemoimmunotherapy trials. Thus, these findings may be difficult to extrapolate in the context of modern systemic therapy.

## 4. Ongoing Trials of LAT in Oligometastatic NSCLC

To further study the role of LAT in oligometastatic disease in various clinical contexts, several international clinical trials are ongoing, including the NRG-L002 (NCT03137771), SARON (NCT02417662), NORTHSTAR (NCT03410043), LONESTAR (NCT03391869), CURB (NCT03808662), STOP (NCT02756793), SABR-COMET 10 (NCT03721341), and ARREST (NCT04530513), among others.

### 4.1. Clinical Trials of LAT in Patients Receiving First-Line Cytotoxic Chemotherapy

Perhaps the largest of these trials is the NRG-L002 (NCT03137771) in the United States—a large Phase II/III randomized, controlled, intergroup study investigating the effect of adding LAT to maintenance systemic therapy in patients with synchronous oligometastatic NSCLC after successful completion of first-line systemic therapy. In this study, LAT can include either SABR or surgery to the metastatic sites. In cases where the primary tumor cannot be safely treated with LAT, 3D-CRT or IMRT to the primary tumor over 3–5 weeks is permitted. Patients are excluded if they have clinical or radiographic evidence of disease progression at the time of study enrollment. Patients are also excluded if they received targeted therapy (i.e., non-cytotoxic chemotherapy) for NSCLC as part of their definitive treatment for the non-metastatic disease at the time of initial diagnosis. The primary endpoints are PFS and OS. As of December 2021, the trial—which opened in 2017—has since completed accrual to the Phase II portion. A planned interim analysis is underway to determine whether to reopen for Phase III. With a final accrual goal of 400 patients, the trial is estimated to complete in August 2027.

In the United Kingdom, the SARON (NCT02417662) trial is a large Phase III multicenter, randomized, controlled clinical study led by the Royal Marsden [51]. Like the NRG-L002 study, the SARON trial aims to assess the effect of LAT (with SBRT only) following standard chemotherapy in patients with a primary tumor and synchronous oligometastatic lesions (i.e., ≤5 lesions in a maximum of 3 organs). Patients are enrolled prior to the initiation of standard chemotherapy, the choice of which is at the discretion of the treating physician but is expected to be in accordance with institutional guidelines. Patients are excluded if they have actionable mutations or >4 brain metastases. The primary outcome is OS. The study, which opened in 2015, is actively enrolling across 21 study locations, aims to accrue 340 patients and is expected to be complete in August 2022.

### 4.2. Clinical Trials of LAT in Patients Receiving Novel Targeted Agents or Immunotherapy

To address the question of safety and efficacy of LAT with targeted therapies and immunotherapy, there are several other ongoing trials that instead aim to assess the impact of LAT within these clinical contexts. For patients with targetable *EGFR* mutations (i.e., exon 19 deletion or *L858R*), the multicenter Phase II NORTHSTAR (NCT03410043) trial aims to study the effect of osimertinib followed by LAT (with either radiation or surgery), followed by adjuvant osimertinib. This will be in comparison to patients who continue osimertinib alone. The primary endpoint is PFS.

Separately, patients without targetable oncogenic driver mutations may be candidates for immunotherapy. Emerging preclinical data on interactions between radiotherapy and the host immune system suggest the existence of synergistic mechanisms by which immunotherapy could be leveraged to improve the efficacy of radiotherapy [52,53]. In addition to its own cytotoxic effects on tumor cells, radiotherapy promotes the recruitment of not only inflammatory (i.e., antigen-presenting cells and cytotoxic CD8+ T cells) but also immunosuppressive cells (i.e., T_reg_ cells). As such, some have hypothesized that this balance, known as radiation-induced tumor equilibrium (RITE), can be tipped towards immune activation via immunotherapies or dual checkpoint blockade [53]. To that end, there are ongoing trials, including the Phase III LONESTAR (NCT03391869) trial, which will test the effect of LAT (also with radiation and/or surgery) following induction and before adjuvant dual-checkpoint blockade with nivolumab and ipilimumab versus dual-checkpoint blockade alone in patients within oligometastatic NSCLC. The primary endpoint is OS. Opened in 2017, the study is expected to complete in December 2022.

### 4.3. Clinical Trials of LAT in Patients with Oligoprogressive Disease

As we continue to refine our understanding of oligometastatic disease from both a biological and technical perspective, we have also come to acknowledge the existence of other unique disease states within the oligometastatic spectrum. A recent ESTRO/ASTRO consensus statement defines oligometastatic disease by the presence of a controlled primary tumor and anywhere between 1 to 5 metastatic lesions, with the caveat that all sites must be safely treatable with LAT [29]. Building on this definition, the ESTRO/EORTC released a comprehensive framework to help better characterize this wide range of disease entities, including but not limited to oligopersistence, oligorecurrence, and, in particular, oligoprogression [54]. In general, oligoprogressive disease refers to the progression of a limited number (1 to 5) of metastatic lesions against the backdrop of otherwise stable polymetastatic or widely metastatic cancer. Although prognosis is likely worse in patients with oligoprogressive (relative to oligometastatic) disease, treatment with aggressive LAT may also play a role in prolonging PFS in this setting.

To that end, there are several trials studying the effect of LAT in oligoprogressive disease. Notable among these is the PROMISE-004: CURB (NCT03808662) trial—a Phase II, randomized, controlled, single-institution study from Memorial Sloan Kettering Cancer Center (MSKCC), which aims to assess the benefit of SBRT to all sites of oligoprogression (up to 5) versus SOC palliative therapy alone [55]. Since the study’s opening in 2017, a total of 107 patients with metastatic breast cancer or NSCLC with progression on at least first-line systemic therapy have been enrolled. Among those with metastatic breast cancer, patients were eligible if they had either triple-negative and/or another high-risk disease (as determined by the treating physician) that has progressed on hormone or systemic therapy, regardless of *ER*, *PR*, or *HER2* hormone receptor status. Among those with metastatic NSCLC, patients were eligible to enroll if they had metastatic NSCLC without an actionable *EGFR* mutation or *ALK/ROS1* rearrangement or if they had an actionable *EGFR* mutation or *ALK/ROS1* rearrangement but developed oligoprogression after treatment with a first-line tyrosine kinase inhibitor. Patients were first stratified by the number of progressive sites (1 vs. 2–5), type of prior systemic therapy (immunotherapy versus other), primary tumor histology, and tumor marker status before randomization.

Early results from a planned interim analysis were presented at ASTRO 2021 [55]. Overall, of the 107 patients enrolled, 102 patients were randomized thus far—including 58 patients with metastatic NSCLC and 44 patients with breast cancer. Approximately 75% of patients had 2 or more sites of oligoprogression, with 47% having >5 sites of metastatic disease. Most patients (86%) with NSCLC did not have an actionable mutation, while 32% of patients with breast cancer were triple negative. After a median follow-up of 51 weeks, there was a PFS benefit in patients who received SBRT compared to standard of care palliation alone (median PFS: 22 weeks vs. 10 weeks, *p* = 0.005)—an effect driven largely by patients treated with SBRT in the NSCLC arm (44 weeks vs. 9 weeks, *p* = 0.004), which remained significant on multivariable Cox analysis (HR 0.38, 95% CI 0.18–77; *p* = 0.007). There was no difference in median PFS in the breast cancer cohort (SBRT, 18 weeks vs. SOC, 17 weeks; *p* = 0.5). This is consistent with the updated results of the NRG-BR002 (NCT02364557) Phase II trial to be presented at ASCO 2022, which found that LAT with SABR did not improve PFS or OS in patients with oligometastatic breast cancer. As such, the study will not be proceeding to the Phase III component [56]. Tsai et al. [55] concluded that based on the findings of this interim analysis, there was a PFS benefit of SBRT to sites of oligoprogressive disease that appeared to differ by histology, thus warranting further investigation.

To date, CURB is the first and only randomized trial to demonstrate the benefit of SBRT alone in oligoprogressive NSCLC. The trial, which is ongoing, is expected to complete in January 2023. However, several other trials investigating the role of LAT in this and other contexts along the oligometastatic spectrum are ongoing. Such as CURB, the randomized Phase II STOP (NCT02756793) trial that aims to study the effect of SABR on all sites of oligoprogressive disease in patients receiving systemic therapy or on observation, with a primary endpoint of PFS. Following the landmark findings of the SABR-COMET trial, which showed a benefit of SABR within the traditional context of oligometastatic disease (≤3 lesions), the SABR-COMET 10 (NCT03721341) trial is a Phase III multicenter study that aims to address the question of safety and efficacy of SABR for patients with a controlled primary tumor and 4–10 metastases. Lastly, although not a randomized study, the Phase I ARREST (NCT04530513) trial is a dose-escalation/de-escalation study that will address important questions regarding the safety of SABR delivered to all sites of polymetastatic disease at five different dose levels, thus providing important data on optimal dose and technique in the oligoprogressive/polymetastatic context.

## 5. Future Directions

### 5.1. Determining Appropriate Patient Selection for LAT in Oligometastatic NSCLC

As randomized evidence continues to emerge in support of LAT across the spectrum of oligometastatic disease, there is a growing need to develop criteria to help guide patient selection. However, prognostication is notoriously difficult and is especially challenging in patients with oligometastatic NSCLC, for whom survival may vary widely due to the biologic heterogeneity associated with different oligometastatic contexts [57]. That said, there have been several attempts to leverage clinical, radiographic, and pathological data to develop predictive and prognostic tools to assess the potential benefit of metastasis-directed LAT.

First, factors associated with survival outcomes in patients with oligometastatic NSCLC have been reported in several meta-analyses: Ashworth et al. [58] validated a model in which synchronous versus metachronous metastases, nodal stage, and adenocarcinoma histology were associated with OS (C-statistic = 0.682). Similarly, Li et al. [59] found that negative nodal status, adenocarcinoma histology, and female gender were favorable predictive factors for OS. With respect to radiographic data, a few retrospective studies found that the incorporation of FDG-PET parameters could also help to improve prognostic modeling: Metabolic tumor volume, total lesion glycolysis, and GLCM energy (gray-level co-occurrence matrix energy) has been shown to be predictive of OS in patients with oligometastatic NSCLC [60,61], whereas maximum standardized uptake value (SUV_max_) and the number of lesions have not [61]. Furthermore, a few predictive nomograms have also been developed in the context of oligometastatic NSCLC. One nomogram model found that KPS, primary tumor histology, primary tumor control, size of the largest metastasis, and the number of metastases (1 vs. >1) were significant prognostic factors for OS after SBRT [62]. A smaller, more recent study showed that the solitary site of metastasis, targetable mutations, intracranial disease, and metachronous oligometastases were all predictive of larger PFS benefits after LAT [63].

With respect to anatomic location, there have been no head-to-head comparisons of clinical outcomes following LAT for intracranial versus extracranial oligometastatic NSCLC, with most single-arm Phase I-II studies of LAT for oligometastatic NSCLC excluding patients with intracranial disease (Table 2). However, in one Phase II single-arm study, 17 out of 39 patients (44%) had a least one brain metastasis treated with LAT, either resection (*n* = 4) or SRS (*n* = 13) [48,64]. Importantly, while 33 out of 39 patients (85%) developed recurrence, only 2 patients had a local recurrence within the treatment field [64]. Among patients with treated brain lesions, 9 out of 17 patients (52%) had recurrent CNS disease; however, none were within the field [64], consistent with prior studies showing excellent local control with SRS [65,66,67,68,69]. Additionally, two of the three Phase II randomized trials included patients with intracranial disease. In the SABR-COMET trial, only three brain lesions were present in the control arm versus only one in the SABR group, thus limiting further interpretation [23,25], while in the Gomez et al. trial, the presence of brain metastases was not significantly associated with OS [22,24]. As such, the current ESTRO-ASTRO consensus statement states that the oligometastatic disease state appears to be largely independent of anatomic location, provided LAT to these sites confers acceptable toxicity [29]. The eligibility criteria of future and ongoing randomized studies should take care to include patients with brain lesions as part of efforts to further elucidate the prognostic implications of intracranial versus extracranial involvement in oligometastatic disease if any.

Another important consideration for patient selection is determining the appropriateness of LAT for oligometastatic NSCLC in the older adult population [70]. Recent estimates suggest that the average age at diagnosis for lung cancer is 70 years old, with most patients being diagnosed at age 65 or older [1]. Notably, the median age of patients in the published randomized Phase II trials ranges between 61 to 69 years in the intervention arm and 63 to 70 years in the control arm; however, eligibility criteria across studies limited enrollment to patients with good performance status (ECOG ≤2 or KPS ≥70) and in the SABR-COMET trial, life expectancy was ≥6 months [21,22,23,24,25]. Moreover, while the incorporation of the above prognostic factors has been helpful in the stratification of patients in existing prospective clinical trials, none currently account for the effect of age or metrics of fitness and frailty outside of performance status alone. As the global population continues to grow older, ongoing and future studies of LAT in oligometastatic NSCLC should consider the incorporation of older adults with lower performance status, as well as the incorporation of fitness and frailty indices into the trial design [70].

### 5.2. Developing Biomarkers for Oligometastatic NSCLC

Lastly, there has been growing interest in the identification and validation of predictive biomarkers of response to local ablative therapy (LAT) in the context of oligometastatic NSCLC. Current genomic analyses have yet to identify robust biomarkers of oligometastatic disease. This is perhaps unsurprising, given the complexity of the numerous signaling pathways involved in metastasis formation and dissemination. As such, it is unlikely that a single driver mutation or pathway alteration exists that would consistently result in an oligometastatic state. Nonetheless, there have been several efforts to characterize epigenetic modifications and tumor microenvironment interactions that underly the oligometastatic phenotype. Preclinical data from Lussier et al. [71] and Uppal et al. [72] showed that the overexpression of microRNAs (miRNAs) in the miRNA-200 family and the 14q32-encoded miRNA cluster may be associated with the development of the oligometastatic phenotype in clinical samples. Lussier et al. went on to validate these findings in a xenograft murine model, demonstrating that the enhancement of the miRNA-200 function results in a significant increase in metastatic burden in a cell line with low metastatic potential [71].

More recently, Tang et al. [73] reported the immune and circulating tumor DNA (ctDNA) correlatives from the MDACC (Gomez et al.) Phase II trial. Of the 49 patients enrolled and randomized in the study, 31 underwent ctDNA analysis, 21 underwent T cell CDR3 variable region sequencing, and 19 had assessments of cytokine concentration. Baseline interleukin 1a was the only cytokine associated with both OS and PFS benefits. Patients who received LAT were found to have changes in T cell clonality, suggesting favorable and early oligoclonal expansion. These patients also had significantly decreased ctDNA burden at early follow-up, compared to the MT/O arm. Although interpretation is limited by small sample size, these findings may suggest that the survival benefit associated with LAT may be mediated by enhanced early immune-mediated tumor killing in oligometastatic NSCLC and that ctDNA could be used post-LAT for the assessment of tumor response and/or early recurrence. However, care should be taken with this approach as levels of ctDNA shedding can vary among patients with intrathoracic only versus extra-thoracic involvement of disease [74].

## 6. Conclusions

With the advent of modern systemic therapies and the output of high-level evidence supporting aggressive local consolidation in the oligometastatic disease setting, the once dismal outlook for patients with Stage IV NSCLC has perhaps never been better. Moreover, the incorporation of oligometastatic NSCLC in the latest iteration of national consensus guidelines [75] marks a growing rec ognition of this distinct disease state by the broader oncology community. Highly anticipated Phase III data from ongoing trials will prove vital to guiding clinical decision-making and treatment recommendations. In the meantime, we must continue endeavors aimed at answering questions regarding appropriate patient selection, optimal timing/technique, and the biological mechanisms underlying the oligometastatic disease state as clinicians begin to adopt these shifting treatment paradigms in routine clinical practice.

## Figures and Tables

**Table 1 cancers-14-03977-t001:** Key definitions of the oligometastatic spectrum.

Key Terms	Definition	References
Oligometastatic disease (OMD)	An intermediate state between local and systemic disease, where radical local treatment of the primary tumor and all metastatic lesions may have curative potential.	[16,29]
	Most studies and current consensus guidelines accept a disease burden of 1–5 lesions, although published randomized Phase II data have only confirmed the benefit in 1–3 lesions thus far, with trials ongoing.	[21,22,23,24,25]
Synchronous OMD	The presence of OMD at the time of (or up to 3 months after) initial diagnosis, with simultaneous detection of the primary tumor and limited metastases.	[32,37]
Metachronous OMD (Also known as “oligorecurrent disease”)	The presence of OMD at least 3 months after initial diagnosis. Most studies stipulate the achievement of primary tumor control as per the definition introduced by Niibe and Hayakwa, or at minimum, prior treatment to the primary with curative intent.	[29,32,37,38]
Oligoprogressive or oligopersistent disease	The progression or persistence of limited (1–5) viable metastases following the receipt of systemic therapy on a background of widely or polymetastatic disease.	[29,39,40]
Polymetastatic disease	The presence of systemic disease, currently defined as >5 metastases, although trials studying the efficacy of LAT for patients with >5 metastases are ongoing (i.e., SABR-COMET 10 (NCT03721341) for 4–10 metastases)	[41,42]

**Table 2 cancers-14-03977-t002:** Phase II single-arm and randomized control trials of LCT in oligometastatic NSCLC.

Study Characteristics		Cohort Characteristics				Treatment Characteristics			Results	
Publication	Design	Population (Primary; Notable Criteria; Sample Size, *N*)	ECOG/KPS Criteria	Age Years, Median (Range)	Disease status (OMD type; EGFR/ALK+; Brain Metastases Treated?)	RT Technique (Dose/Fx)	Surgery as LCT?	Systemic Therapy?	Endpoints	Clinical Outcomes
**Salama et al.** *Clin Cancer Res* 2008 [45]	Dose escalation trial, single-arm	Multiple (18% NSCLC) *N* = 61	ECOG ≤ 2, Life expectancy >3 months	NR	Synchronous; NR; No brain mets treated	SBRT (24–48 Gy/3fx)	No	Yes (80.3% pre- RT)	**Primary:** Dose-limiting toxicities **Secondary:** PFS, OS	**Median F/U:** 29 months **Median PFS:** 5.1 months **1y OS:** 81.5%, **2y OS:** 56.7% Acute G3+ toxicity (*n* = 2) Late G3+ toxicity (*n* = 6)
**Iyengar et al.** *JCO* 2014 [46]	Phase II, single arm	NSCLC, ≤6 metastases (≤3 in lungs/liver), *N* = 24	KPS ≥ 70	67 (56–86)	Oligoprogressive; 0/13 EGFR/ALK+; No brain mets treated	SABR (19–20 Gy/1fx, 35–40 Gy/5fx, 27–33 Gy/3fx)	No	Concurrent Erlotinib (100%)	**Primary:** PFS **Secondary:** OS, toxicity	**Median F/U:** 16.8 (range, 3.4–60.3) months **6mo PFS:** 69%, **Median PFS:** 14.7 months **Median OS:** 20.4 months All G3+ toxicity, 8%
**Colleen et al.** *Ann Oncol* 2014 (NCT01185639) [47]	Phase II, single arm	NSCLC primary (controlled), ≤5 lesions on PET, *N* = 26	WHO ≤ 2	62 (47–75)	Synchronous (73%), Metachronous (27%); 2/26 EGFR/ALK+; No brain mets treated	SBRT (50 Gy/10fx)	No	Induction chemotherapy (65%)	**Primary:** PFS, OS **Secondary:** toxcity	**Median F/U:** 16.4 (3–40) months **1y PFS:** 45%, **Median PFS:** 11.2 months **1y OS:** 67%, **Median OS:** 23 months G3+ pulmonary toxicity, 8%
**De Ruysscher et al.***JTO* 2012, 2018 (NCT01282450) [48]	Phase II, single arm	NSCLC primary, <5 metastases, *N* = 39	WHO ≤ 2	62 (44–81)	Synchronous; 33% EGFR/ALK+; **Brain mets treated**	Conventional SBRT, SRS (Various)	Yes	SOC maintenance chemotherapy	**Primary:** OS **Secondary:** PFS, toxicity	*Long-term results* **Minimum F/U:** 7 years **Median PFS:** 12.1 (95% CI 9.6–14.3) months **Median OS:** 13.5 (95% CI 7.6–19.4) months G3+ toxicity, 18%
**Petty et al.** *IJROBP* 2018 (NCT011856639) [49]	Phase II, single arm	NSCLC, ≤5 metastases (across ≤3 sites other than mediastinal/hilar nodes) *N* = 27	ECOG ≤ 2	65 (49–83)	Synchronous and metachronous; Excluded; No brain mets treated	Conventional SBRT, SRS (Various)	No	Induction chemotherapy	**Primary:** PFS **Secondary:** OS, toxicity	**Median F/U:** 24.1 months **Median PFS:** 11.2 (7.6–15.9) months **Median OS:** 28.4 (14.5–45.8) months G3+ toxicity, 0%
**Bauml et al.** *JAMA Oncol* 2019 (NCT02316002) [50]	Phase II, single arm	NSCLC, ≤4 metastases, *N* = 51	ECOG ≤ 1	64 (46–82)	Synchronous (69%), Metachronous (31%); NR; No brain mets treated	Conventional SBRT (NR)	Yes	Pembrolizumab	**Primary:** PFS **Secondary:** OS, toxicity	**Median F/U:** 25 months **Median PFS:** 19.1 (95% CI 9.4–28.7) months **1y OS:** 90.4%, **2y:** 77.5%, **Median OS:** 41.6 months G2-4 pneumonitis, 11%
**Gomez et al.** *Lancet Oncol* 2016 *JCO* 2019 (NCT01725165) [24]	Phase II RCT	NSCLC primary ≤3 metastases without progression after 3 months of systemic therapy, *N* = 49	ECOG ≤ 2	**MT/O:** 61 (43–80) **LCT:** 63 (43–83)	Synchronous; 8/49 EGFR/ALK+; **Brain mets treated**	Conventional SBRT, SRS (Various)	Yes	SOC maintenance chemotherapy	**Primary:** PFS **Secondary:** OS, toxicity, appearance of new lesions	**MT/O vs. LCT (long-term results)** **Median PFS:***p* = 0.022, 4.4 (95% CI, 2.2 to 8.3) vs. 14.2 months (95% CI, 7.4 to 23.1) **Median OS:** *p* = 0.017, 17.0 (95% CI, 10.1 to 39.8) vs. 41.2 months (18.9 to not reached) Longer survival after progression for the LCT group. No additional toxicity ≥G3 observed.
**Iyengar et al.** *JAMA Onc* 2018 (NCT02045446) [21]	Phase II RCT	NSCLC primary, ≤5 metastases with SD after induction, *N* = 29	KPS ≥ 70	**Maintenance chemotherapy:** 63.5 (51–78) **SABR before maintenance:** 70 (51–79)	Synchronous; Excluded; No brain mets treated	SABR (21–27 Gy/1fx, 26.5–33 Gy/3fx, 30–37.5 Gy/5fx, 45 Gy/15fx)	No	SOC maintenance chemotherapy	**Primary:** PFS **Secondary:** In-field local control, out-of-field disease progression, safety, OS	**Maintenance chemotherapy vs. SABR** **Median PFS:***p* = 0.01, 3.5 vs. 9.7 months (HR 0.304, 95% CI 0.113–0.815) Toxicity similar in both arms. No in-field failures, fewer recurrences in SABR arm.
**Palma et al.** SABR-COMET *Lancet* 2019 *JCO* 2020 (NCT01446744) [23]	Phase II RCT	Multiple primary types, ≤5 metastases *N* = 99	ECOG 0-1 Life expectancy ≥ 6 months	**SOC:** 69 (64–75) **SABR:** 67 (59–74)	Synchronous; NR; **Brain mets treated**	SABR, SRS (30–60 Gy/3-8fx, 16–24 Gy/1fx)	No	SOC maintenance chemotherapy	**Primary:** OS **Secondary:** QOL, toxicity, PFS, lesional control (LC) rate, number of cycles of further chemotherapy/systemic therapy	**SOC vs. SABR (long-term)** **5-year OS:***p* = 0.006, 17.7 (95% CI 6% to 34%) vs. 42.3 months (95% CI 25%-56%). **5-year PFS:** *p* = 0.001, not reached (95% CI 0% to 14%) vs. 17.3% (95% CI 8% to 30%) No differences in QOL between arms.

**Abbreviations:** ECOG, Eastern Cooperative Oncology Group; Fx, fractions; F/U, follow-up; KPS, Karnofsky Performance Status; LCT, local consolidative therapy; NSCLC, non-small cell lung cancer; NR, not reported; OMD, oligometastatic disease; OS, overall survival; PFS, progression-free survival; QOL, quality-of-life; RCT, randomized control trial; RT, radiotherapy; SABR, stereotactic ablative radiotherapy; SBRT, stereotactic body radiotherapy; SRS, stereotactic radiosurgery; SOC, standard-of-care; WHO, World Health Organization.

## Data Availability

Not applicable.

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
