# Peer review of "Local Consolidative Therapy for Oligometastatic Non-Small Cell Lung Cancer"

_cancers, 2022, doi:10.3390/cancers14163977_

Round 1

Reviewer 1 Report

The present manuscript reports a comprehensive overview on local consolidative therapies for oligometastatic NSCLC.

Some comments:

  • The manuscript is well written and updated. However, the inclusion of a table summarizing the critical aspects/major findings of the studies discussed might be useful.
  • The identification of predictive biomarkers is a critical issue in this setting. The use of ctDNA monitoring is a promising strategy. However, ctDNA shedding varies significantly between patients with intra- ed extra-thoracic involvement (Cortinovis D, et al. Transl Lung Cancer 2021). These considerations should be included in paragraph 4.
  • Discussing the role of LAT in immunotherapy-treated patients, the authors should provide some additional details on the preclinical rational of this approach (Weichselbaum RR, et al. Nat Rev Clin Oncol 2017).
  • As the definition of oligo-metastatic NSCLC significantly varied in most of the studies reported so far, a table with the definitions might be useful.

Author Response

Reviewer 1:

The present manuscript reports a comprehensive overview on local consolidative therapies for oligometastatic NSCLC.

Some comments:

The manuscript is well written and updated. However, the inclusion of a table summarizing the critical aspects/major findings of the studies discussed might be useful.

Thanks very much for the positive feedback. We have included a summary of all pertinent single arm and randomized Phase II trials to date in Table 2.

The identification of predictive biomarkers is a critical issue in this setting. The use of ctDNA monitoring is a promising strategy. However, ctDNA shedding varies significantly between patients with intra- ed extra-thoracic involvement (Cortinovis D, et al. Transl Lung Cancer 2021). These considerations should be included in paragraph 4.

We have included a consideration regarding differential ctDNA shedding in patients with intra- and extra-thoracic involvement as follows:

“…Although interpretation is limited by a small sample size, these findings may suggest that the survival benefit associated with LAT may be mediated by enhanced early immune mediated tumor killing in oligometastatic NSCLC, and that ctDNA could be used post-LAT for assessment of tumor response and/or early recurrence. However, care should be taken with this approach as levels of ctDNA shedding can vary from patients with intrathoracic only versus extra-thoracic involvement of disease (Cortinovis et al. Transl Lung Cancer 2021).”

Section 5.2, Paragraph 2, Page 9

Discussing the role of LAT in immunotherapy-treated patients, the authors should provide some additional details on the preclinical rationale of this approach (Weichselbaum RR, et al. Nat Rev Clin Oncol 2017).

We have included a paragraph on the preclinical rationale behind LAT in immunotherapy patients, including information from the Weichselbaum et al. paper as follows:

“…Emerging preclinical data on interactions between radiotherapy and the host immune system, suggest the existence of synergistic mechanisms by which immunotherapy could be leveraged to improve the efficacy of radiotherapy (Liang et al. J Immunol 2013, Weichselbaum et al. Nat Rev Clin Oncol 2017). In addition to its own cytotoxic effects on tumor cells, radiotherapy promotes the recruitment of not only inflammatory (i.e., antigen presenting cells and cytotoxic CD8+ T cells) but also, immunosuppressive cells (i.e., Treg cells). As such, some have hypothesized that this balance, known as radiation-induced tumor equilibrium (RITE), can be tipped towards immune activation via immunotherapies or dual checkpoint blockade (Weichselbaum et al. Nat Rev Clin Oncol 2017).. To that end, there are ongoing trials, including the Phase III LONESTAR (NCT03391869) trial, which will test the effect of LAT (also with radiation and/or surgery) following induction and before adjuvant dual-checkpoint blockade with nivolumab and ipilimumab versus dual-checkpoint blockade alone in patients with in oligometastatic NSCLC.”

Section 4.2, Paragraph 1, Page 7

As the definition of oligo-metastatic NSCLC significantly varied in most of the studies reported so far, a table with the definitions might be useful.

We have included a summary of various oligometastatic disease states in Table 1.

Reviewer 2 Report

A very well written and well-focused review.

minor suggestions:

  • a small paragraph on the role aggressive therapy to the primary site (not only the metastases) would be welcome
  • a comment on the role of the location of the metastases would be welcome. Once, intracranial disease is mentioned to be predictive for larger PFS, is this in line with other data?

Typos?:

  • When LAT is first defined, please use the phrase local ablative therapy to explain this
  • in Section 4.2. after Lussier et al. there is timestamp instead of a reference?
  • in Line 176 "had a significantly OS" -- maybe "higher" or "improved" is missing

Author Response

Reviewer 2:

A very well written and well-focused review. Minor suggestions:

A small paragraph on the role aggressive therapy to the primary site (not only the metastases) would be welcome.

Thank you for the positive comments. We have included two paragraphs detailing the differences between synchronous versus metachronous oligometastatic disease, as well as a table summarizing these definitions (Table 1). We also included a short discussion on the role of aggressive therapy to the primary site for the purposes of primary local control in patients with synchronous oligometastatic disease as follows:

In a recent ESTRO-ASTRO consensus statement published in 2020, Lievens et al. defined the oligometastatic disease state primarily by the existence of 1-5 metastatic lesions, largely independent of location, provided that they are all safely treatable (Table 1). Notably, primary tumor control is considered optional (Lievens et al. Radiother Oncol 2020). As such, oligometastatic disease can be defined as either synchronous or metachronous with diagnosis of the primary tumor. Synchronous oligometastatic disease is detected at the time of and up to 3 months after initial diagnosis, with the patient having an untreated primary tumor and limited metastases simultaneously (Lievens et al. Radiother Oncol 2020). In contrast, metachronous oligometastatic disease (sometimes referred to as “oligorecurrent” disease), is defined as detection of limited metastases at least 3 months after the initial diagnosis, and often after control of the primary tumor is achieved after treatment with curative intent (Lievens et al. Radiother Oncol 2020).

Importantly, authors noted that the distinction between synchronous versus metachronous oligometastatic disease is critical as some studies report a better prognosis for among patients with metachronous oligometastatic disease (Franzese et al. J Urol 2019, Fode et al. Radiother Oncol 2015), though this has not been consistently observed (Fleckenstein et al. BMC Cancer 2016, Sharma et al. Acta Oncol 2019). Additionally, while there are some concerns that shorter disease-free interval between attainment of primary tumor control and subsequent development of metachronous oligometastatic disease may also be of prognostic significance (Park et al. Anticancer Research 2015, Oh et al. Acta Oncol 2012, Sutera et al. IJROBP 2019), limited data preclude a consensus regarding its importance.

Section 2.1, Paragraphs 1-2, Page 3

A comment on the role of the location of the metastases would be welcome. Once, intracranial disease is mentioned to be predictive for larger PFS, is this in line with other data?

We have included a paragraph discussing the role of anatomic location, with special attention to data on LAT for oligometastatic NSCLC with intracranial versus extracranial disease involvement as follows:

With respect to anatomic location, there have been no head-to-head comparisons of clinical outcomes following LAT for intracranial versus extracranial oligometastatic NSCLC, with most single arm Phase I-II studies of LAT for oligometastatic NSCLC excluding patients with intracranial disease (Table 2). However, in one Phase II single arm study, 17 out of 39 patients (44%) had a least one brain metastasis treated with LAT, either resection (n=4) or SRS (n=13) (De Ruysscher et al. JTO 2012). Importantly, while 33 out of 39 patients (85%) developed recurrence, only 2 patients had a local recurrence within the treatment field (De Ruysscher et al. JTO 2012). Among patients with treated brain lesions, 9 out of 17 patients (52%) had recurrent CNS disease; however, none were within field (De Ruysscher et al. JTO 2012), consistent with studies showing excellent local control with SRS (Aoyama et al. JAMA 2006, Chang et al. Lancet Oncol 2009, Kocher et al. JCO 2011, Gray et al. Lung Cancer 2014, Brown et al. JAMA 2016, Mazzola et al. Front Oncol 2019). Additionally, two of the three Phase II randomized trials included patients with intracranial disease. In the SABR-COMET trial, only 3 brain lesions were present in the control arm versus only 1 in the SABR group, thus limiting further interpretation (Palma et al. Lancet 2019, JCO 2020), while in the Gomez et al. trial, the presence of brain metastases was not significantly associated with OS (Gomez et al. Lancet Oncol 2016, JCO 2019). As such, the current ESTRO-ASTRO consensus statement states that the oligometastatic disease state appears to be largely independent of anatomic location, provided LAT to these sites confers acceptable toxicity (Lievens et al. Radiother Oncol 2020)). Eligibility criteria of future and ongoing randomized studies should take care to include patients with brain lesions as part of efforts to further elucidate the prognostic implications of intracranial versus extracranial involvement in oligometastatic disease, if any.

Section 5.1, Paragraph 3, Page 9

Typos?

  • When LAT is first defined, please use the phrase local ablative therapy to explain this in Section 4.2.
  • After Lussier et al. there is timestamp instead of a reference?
  • In Line 176 "had a significantly OS" -- maybe "higher" or "improved" is missing

We have addressed and corrected all typos identified above.